# A global change in RNA polymerase II pausing during the *Drosophila* midblastula transition

Kai Chen[1], Jeff Johnston[1], Wanqing Shao[1], Samuel Meier[1], Cynthia Staber[1], Julia Zeitlinger[1,2]*

[1]Stowers Institute for Medical Research, Kansas City, United States; [2]Department of Pathology, University of Kansas Medical Center, Kansas City, United States

**Abstract** Massive zygotic transcription begins in many organisms during the midblastula transition when the cell cycle of the dividing egg slows down. A few genes are transcribed before this stage but how this differential activation is accomplished is still an open question. We have performed ChIP-seq experiments on tightly staged *Drosophila* embryos and show that massive recruitment of RNA polymerase II (Pol II) with widespread pausing occurs de novo during the midblastula transition. However, ~100 genes are strongly occupied by Pol II before this timepoint and most of them do not show Pol II pausing, consistent with a requirement for rapid transcription during the fast nuclear cycles. This global change in Pol II pausing correlates with distinct core promoter elements and associates a TATA-enriched promoter with the rapid early transcription. This suggests that promoters are differentially used during the zygotic genome activation, presumably because they have distinct dynamic properties.

*For correspondence: jbz@stowers.org

**Competing interests:** The authors declare that no competing interests exist.

**Reviewing editor**: Ruth Lehmann, New York University School of Medicine, United States

## Introduction

The development of a fertilized egg is initially under the control of maternal products and then becomes under zygotic control when transcription begins. In animals such as *Xenopus*, zebrafish and *Drosophila*, development begins with rapid, synchronous cell divisions without gap phases (*Lamb and Laird, 1976*; *Newport and Kirschner, 1982*; *Tadros and Lipshitz, 2009*). During the midblastula transition (MBT), cells switch to prolonged asynchronous divisions and this coincides with a massive increase in zygotic transcription. Before the MBT, only a small set of 'pre-MBT genes' are expressed. How this differential transcription is set up globally during these early stages of development, including the role of histone modifications and the recruitment of RNA Polymerase II (Pol II), has been the subject of considerable interest (*Akkers et al., 2009*; *Vastenhouw et al., 2010*; *Lindeman et al., 2011*). In mammalian embryonic stem cells, as well as early *Drosophila* embryos, paused Pol II is frequently found at developmental control genes (*Guenther et al., 2007*; *Zeitlinger et al., 2007*; *Min et al., 2011*), but it is not known when pausing is first established in the embryo. Pol II pausing prior to activation may promote the rapid and synchronous induction of genes (*Boettiger and Levine, 2009*; *Adelman and Lis, 2012*), but it is unclear whether Pol II can be recruited and paused during the rapid early nuclear cycles prior to the MBT (*Kim and Jinks-Robertson, 2012*; *Petruk et al., 2012*).

## Results

### Massive de novo Pol II recruitment during the midblastula transition

We first probed the status of Pol II in the early embryo by immunostainings (*Figure 1A*). In *Drosophila*, the MBT mainly occurs in the interphase of nuclear cycle 14, just before cellularization and subsequent gastrulation (*Foe et al., 1993*), although there is some evidence that this may already occur in nuclear

**eLife digest** Fertilized eggs—zygotes—develop into embryos via several distinct stages. In many animals, the zygote initially undergoes rapid rounds of genome replication; however, this hectic activity is not controlled by the zygote itself. Instead, the mother deposits RNA molecules in the egg as it forms inside her, and after the egg has been fertilized, these RNA molecules are translated into proteins that guide the development of the early embryo. Only at a stage called midblastula transition does the zygote take over control by transcribing its own RNA molecules.

Fruit flies start to transcribe their own genes en masse after completing thirteen rounds of DNA replication. However, some genes are already transcribed during the rapid cycles of DNA replication earlier in development. How these early genes are transcribed, and how the embryo shifts to more widespread transcription during the midblastula transition, are not well understood. In particular, it is not known if the molecular machinery needed to transcribe the genes is recruited a long time before transcription starts, or if it is recruited 'just in time'. Here, Chen et al. explore how genes are switched on in the fruit fly zygote.

Genes are transcribed by a protein complex called RNA polymerase, which binds to DNA sequences, called promoters, within the genes. Chen et al. used a technique called ChIP-Seq to determine how much RNA polymerase was bound to the DNA before, during and after the midblastula transition. Before the transition—from about eight rounds of DNA replication onward—RNA polymerase was bound to only about 100 genes, and was active in most of these cases. In contrast, after the transition, RNA polymerase had been recruited to the promoters of around 4000 genes (fruit flies have a total of about 14,000 genes). However, it was often found in a paused, rather than active, form, at these genes, which is thought to help ensure that their transcription can occur on a precise schedule.

Chen et al. then used computer analyses to test the theory that differences in the DNA sequences of the gene promoters might determine which genes the RNA polymerase bound to, and whether or not the polymerase underwent pausing or became active immediately. Strikingly, there were clear differences in the sequence motifs that recruited RNA polymerase to the promoters of genes that were transcribed immediately and those that showed pausing of the polymerase. Moreover, genes that were transcribed before the midblastula transition were shorter, on average, than those transcribed after. This suggests that transcription during the rapid genome replication cycles has to occur quickly and therefore lacks pausing. Together, these findings present a biological rationale for differences in how genes are first transcribed during fruit fly development.

cycle 13 (*Harrison et al., 2011*). Before the MBT, a small fraction of genes (*De Renzis et al., 2007*; *ten Bosch et al., 2006*) may be transcribed as early as nuclear cycle 8.

Our immunostainings show that unphosphorylated Pol II can be detected in nuclei from the earliest cleavage stages on, thus before the beginning of transcription. However, both TATA-box binding protein (TBP), the key subunit of TFIID, which binds to the promoter prior to Pol II recruitment, as well as Serine 5-phosphorylation of the C-terminal domain of Pol II (Ser5-P Pol II), which marks transcriptional initiation, can only be first detected during nuclear cycles 8–12, when significant transcription of pre-MBT genes occurs. This was the first indication that Pol II is recruited to promoters de novo during the zygotic genome activation.

We next performed chromatin immunoprecipitation experiments coupled to deep sequencing (ChIP-seq) to analyze the occupancy of Pol II, TBP and histone modifications in pre-MBT embryos (nuclear cycles 8–12), MBT embryos (nuclear cycles 13–14), as well as post-MBT embryos as control (*Figure 1B*). Although the large amount of *Drosophila* embryos required for ChIP-seq can be collected by conventional means, such collections always contain a fraction (5–20%) of older embryos due to maternal egg holding and thus cannot be used to study very early stages of embryogenesis (*Harrison et al., 2011*). To eliminate this contamination, we stained our embryo collections with DAPI and removed 'out-of-stage' embryos under a microscope with a pipette (*Figure 1B*, *Figure 1—figure supplement 1*, and see 'Materials and methods').

The ChIP-seq data from these hand-sorted embryos have robust and reproducible signals in replicates (*Figure 1—figure supplement 2A,B*). Despite the high Pol II signal in the pre-MBT sample, Pol II only

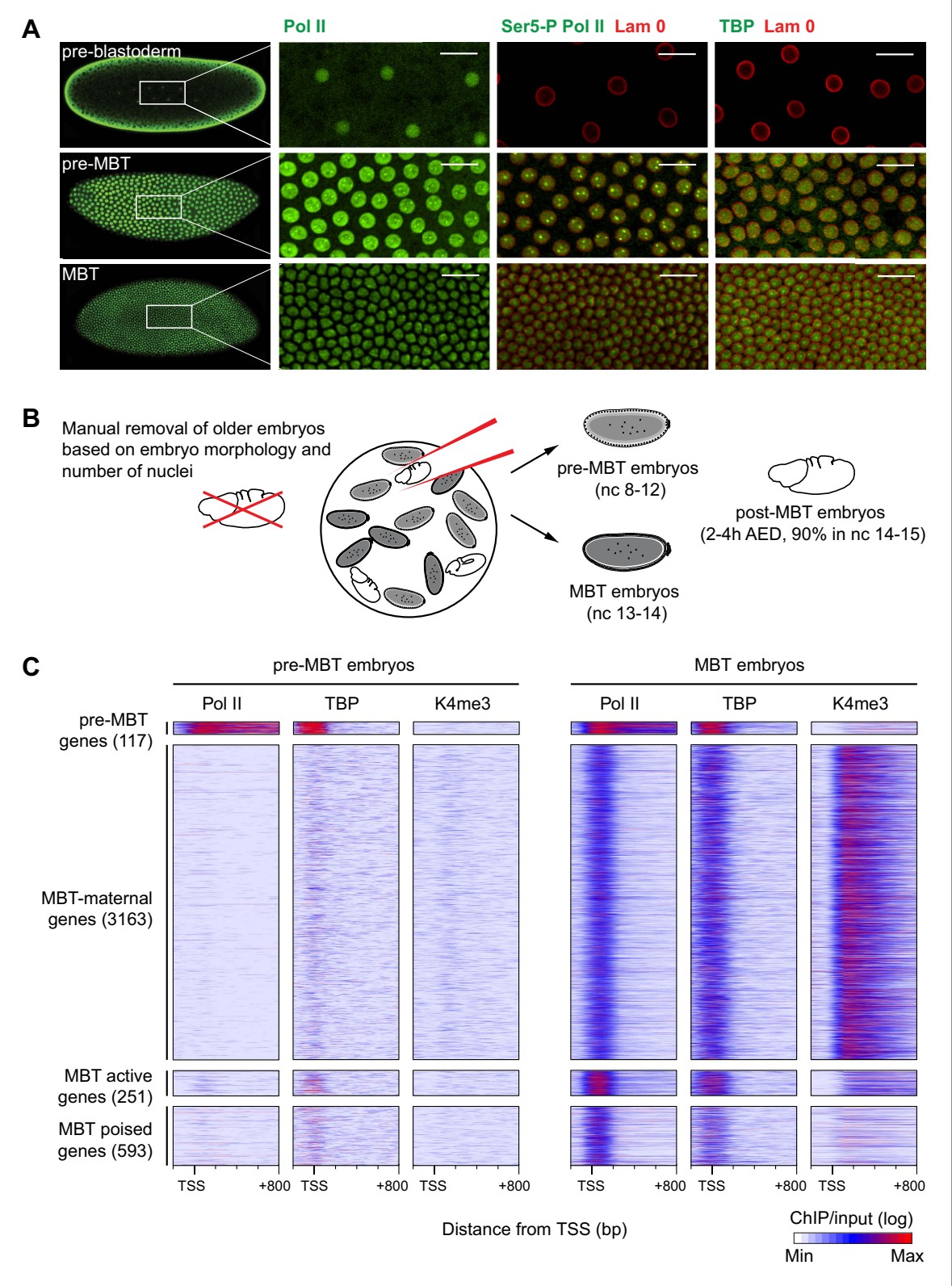

**Figure 1**. Global recruitment of Pol II during the Drosophila midblastula transition. (**A**) Immunostainings of embryos during pre-blastoderm stages (nc 1–7), pre-MBT (nc 8–12) and MBT (nc 13–14) suggest that the initiated form of Pol II (serine-5-phosphorylation of the CTD repeats—Ser5-P), as well as TBP are only detectible in the nuclei (outlined by the Lam 0 in red) of pre-MBT embryos when zygotic transcription begins (scale = 20 μm). (**B**) Outline of *Figure 1. Continued on next page*

*Figure 1. Continued*

the hand-sorting of embryo collections for ChIP-Seq experiments. (**C**) Heat map of ChIP-seq enrichments across all genes that are significantly bound by Pol II during MBT. Pre-MBT genes are also significantly bound in the pre-MBT sample; MBT-maternal genes also have maternally provided transcripts in the early embryo (RPKM > 1 during nc 10); the remaining genes are MBT-zygotic genes. Among the latter group, MBT active genes are expressed during the MBT (RPKM > 5 at nc 14D), while the transcript levels of MBT poised genes are below this threshold. Each line shows the normalized enrichments for a gene from −200 bp to +800 bp from the TSS. Note that Pol II is only bound to few pre-MBT genes before the MBT and that there is massive de novo recruitment of Pol II during the MBT. AED = after egg deposition, nc = nuclear cycle.

The following figure supplements are available for figure 1:

**Figure supplement 1**. Standards for staging pre-MBT and MBT embryos.

**Figure supplement 2**. High reproducibility of pre-MBT and MBT Pol II ChIP-seq data and agreement with previous mRNA data.

**Figure supplement 3**. ChIP-seq occupancy of Pol II and TBP at pre-MBT genes with complex patterns. The reads were normalized to the total read count.

**Figure supplement 4**. Pre-MBT genes identified by ChIP-Seq show high conservation scores among insect genomes.

occupies around a hundred genes before the MBT (*Figure 1C*, Supplementary file hosted by Dryad [7.6 Mb; *Chen et al., 2013*]). These genes include previously described pre-MBT genes, as defined by in situ hybridization (*ten Bosch et al., 2006*) and microarray data (*De Renzis et al., 2007*) (*Figure 1—figure supplement 2C*). In contrast, Pol II and TBP are recruited de novo to 4007 promoters during the MBT, which equates to roughly a third of all genes (*Figure 1C*, Supplementary file hosted by Dryad [7.6 Mb]). This shows that there is massive de novo recruitment of Pol II during the MBT.

## No apparent Pol II pausing at the earliest transcribed genes

To obtain a complete list of pre-MBT genes occupied by Pol II before the MBT, we identified all genes with at least twofold enrichment of Pol II over input at the transcription start site across four Pol II ChIP-seq replicates. From this list, we removed 12 genes that were likely false positives as a result of Pol II read-through from a nearby gene and added 10 genes that were missed due to un-annotated alternative start sites (examples in *Figure 1—figure supplement 3A,B,C*). This yielded 117 pre-MBT genes, many of which have known functions in sex determination, cellularization, anterio-posterior patterning and dorso-ventral patterning (*Table 1*). Among them are also 14 precursors of non-coding RNAs, which are involved in maternal RNA degradation, dosage compensation, and RNA splicing, as well as many genes whose function is unknown but that are well conserved among insect species (*Figure 1—figure supplement 4*, Supplementary file hosted by Dryad [7.6 Mb]).

Inspection of the Pol II occupancy revealed that most pre-MBT genes have no notable enrichment of Pol II at the pause site (+30–50 bp downstream from the transcription start site [*Zeitlinger et al., 2007*]) when they are initially transcribed, while TBP occupancy is found upstream (on average −20 bp from the transcription start site) (*Figure 2*). When quantifying the degree of pausing with the pausing index (see 'Materials and methods'), pre-MBT genes are indeed much less paused than MBT genes (*Figure 2A*). However, a small number of pre-MBT genes have a higher pausing index, lack enrichment of Pol II within the gene body and thus may be paused.

We then classified the pre-MBT genes based on their Pol II occupancy during development and identified three distinct groups (*Figure 2B,C*, Supplementary file hosted by Dryad [7.6 Mb]). Genes in the first group ('pre-MBT not-paused', n = 77) have highest expression during cellularization (nuclear cycle 14) and tend to diminish in expression thereafter (*Figure 2B*). These genes appear never to become paused during early development (see *SNCF* and the average profile in *Figure 2C*). Genes in the second group ('pre-MBT dual', n = 30) initially show little evidence of pausing; however, higher levels of Pol II gradually accumulate at the pause site during and after the MBT (see *ac* and the average profile in *Figure 2C*). Finally, there is a small group of genes ('pre-MBT paused', n = 10) that appear to have Pol II that is paused or non-productive even at the pre-MBT stages (see *sim* and the average profile in *Figure 2C*). This is consistent with the expression of these genes; their transcript levels rise much later during pre-MBT stages as compared to the first two groups (*Figure 2B*). This suggests that Pol II pausing exists during pre-MBT stages but that most genes are non-paused.

**Table 1.** Classification of pre-MBT genes

| Function | Gene names |
| --- | --- |
| Sex determination | **Dpn**, sisA, Sxl, os |
| Cellularization | nullo, Sry-alpha, kuk, bnk, slam |
| Anterio-posterior patterning | cad, hb, gt, kni, tll, eve, h, run, slp1, odd, ftz, *Egfr* |
| Dorso-ventral patterning | **sna**, **esg**, **Nrt**, **glec**, **ac**, **l(1)sc**, **Tom**, **BobA**, **m4**, zen, zen2, tsg, tld, scw, Neu2, sc, fd19B, bnb, Bro, Brd, Ocho, amos, ato, *sim*, *lea* |
| Other function | **Taf4**, **wech**, **Corp**, **toc**, **spri**, Z600, halo, SNCF, CG4570, spo, hrg, *sca*, *Lac*, *RpL3*, *btsz*, *αTub84B* |
| Non-coding RNA | mir-9a, mir-309, roX1, snRNA:U5:34A, snRNA:U4atac:82E, snRNA:U1:82 Eb, snRNA:U5:23D, snRNA:U5:38ABb, snRNA:U5:14B, snRNA:U4:38AB, snRNA:U1:95Cc |
| Unknown function | |
| Localized expression | **gk**, **CG9894**, **CG5059**, sala, term, CG14427, CG8960, CG13711, CG13713, CG15876, CG6885, CG7271, CG14014 |
| Ubiquitous expression | Bsg25A, Bsg25D, CG15634, CG15382 |
| Others | **CG2201**, **CG42666**, CG43659, CG13716, CG13712, CG13000, CG13465, CG14561, CG18269, CG14915, CG16813, CG15479, CG15480, CG4440, CG14317, CG13427, CG34137, CG34214, CG34224, CG34266, CG16815, CG42762, CG43184, CG 9775, CG9883, CR43887, *CG9821*, CG33232 |

Bold marks the pre-MBT dual genes and italic marks the pre-MBT paused genes.

As a control, we analyzed Pol II pausing at genes that are newly occupied by Pol II during the MBT ('MBT genes', Supplementary file hosted by Dryad [7.6 Mb]). We first subtracted from them the 3163 genes that are also maternally expressed ('MBT-maternal genes') because they are known to be enriched for broadly expressed housekeeping genes (*Rach et al., 2009*). The remaining 844 genes ('MBT-zygotic genes') frequently have high Pol II occupancy at the pausing site and a high pausing index (*Figure 3*), suggesting that Pol II pausing is widespread during the MBT.

When we analyzed the expression of these MBT-zygotic genes (*Lott et al., 2011*), we found that 251 genes (30%) are expressed at significant levels during late nuclear cycle 14 ('MBT active genes', see *brk* and the average profile in *Figure 3A*), while the remaining 593 genes ('MBT poised genes', see *Dr* and the average profile in *Figure 3A*) are expressed at very low levels typical for paused genes poised for activation (*Zeitlinger et al., 2007*; *Adelman and Lis, 2012*). This difference in expression was confirmed by analyzing their in situ hybridization patterns: while the MBT active genes are expressed very early, the MBT poised genes tend to be first detected at later embryonic stages (stages 9–10 or later) (*Figure 3B*). Thus, many genes become newly paused during the MBT and are poised for later activation.

The significant difference seen in Pol II pausing between pre-MBT and MBT genes is likely to be biologically meaningful. Genes expressed before the MBT have to be transcribed particularly fast because the nuclear cycle is extremely short (8 min in nuclear cycle 10, increasing to 13 min in nuclear cycle 12 [*Foe et al., 1993*]) and progression through mitosis causes abortion of nascent transcripts (*Rothe et al., 1992*; *Shermoen and O'Farrell, 1991*). As previously noted (*De Renzis et al., 2007*), we also found that the pre-MBT genes are particularly short (median of 1228 bp vs 6024 bp, Mann–Whitney test p<10$^{-20}$) and more often intronless (54.6% vs 9.2%, Fisher test p<10$^{-23}$) compared to MBT-zygotic genes. We also noticed that pre-MBT genes frequently use the transcription start site that yields the shortest transcript (62.9% vs 30.6%, Fisher test p<0.0003) (*Table 2*, and examples in *Figure 1—figure supplement 3C,D*). This supports the idea that rapid transcription is important during the fast cleavage cycles before the MBT and makes it plausible that the lack of Pol II pausing is advantageous for pre-MBT transcription.

## Absence of H3K27me3 and H3K4me3 before transcription

We next explored potential mechanisms that could explain the difference between pre-MBT and MBT genes. Histone modifications such as H3K4me3 and H3K27me3 are present in embryonic stem cells (*Bernstein et al., 2006*), in human sperm (*Hammoud et al., 2009*), and may be present in zebrafish embryos before gene activation (*Vastenhouw et al., 2010*; *Lindeman et al., 2011*). However, we did not find these marks in *Drosophila* nuclei prior to gene activation. In both immunostainings and ChIP-seq data, the signal of H3K4me3, a modification associated with gene activation, only starts to be detectable during the MBT (*Figure 4A*, see also the ChIP-seq results in *Figure 1C*). H3K27me3, a marker of Polycomb-mediated gene silencing and possibly epigenetic memory, can be detected in nuclei and in polar bodies at the earliest cleavage stages but is then undetectable in somatic nuclei until after the MBT (*Figure 4C*). Consistent with this, the ChIP-seq H3k27me3 signal at Polycomb

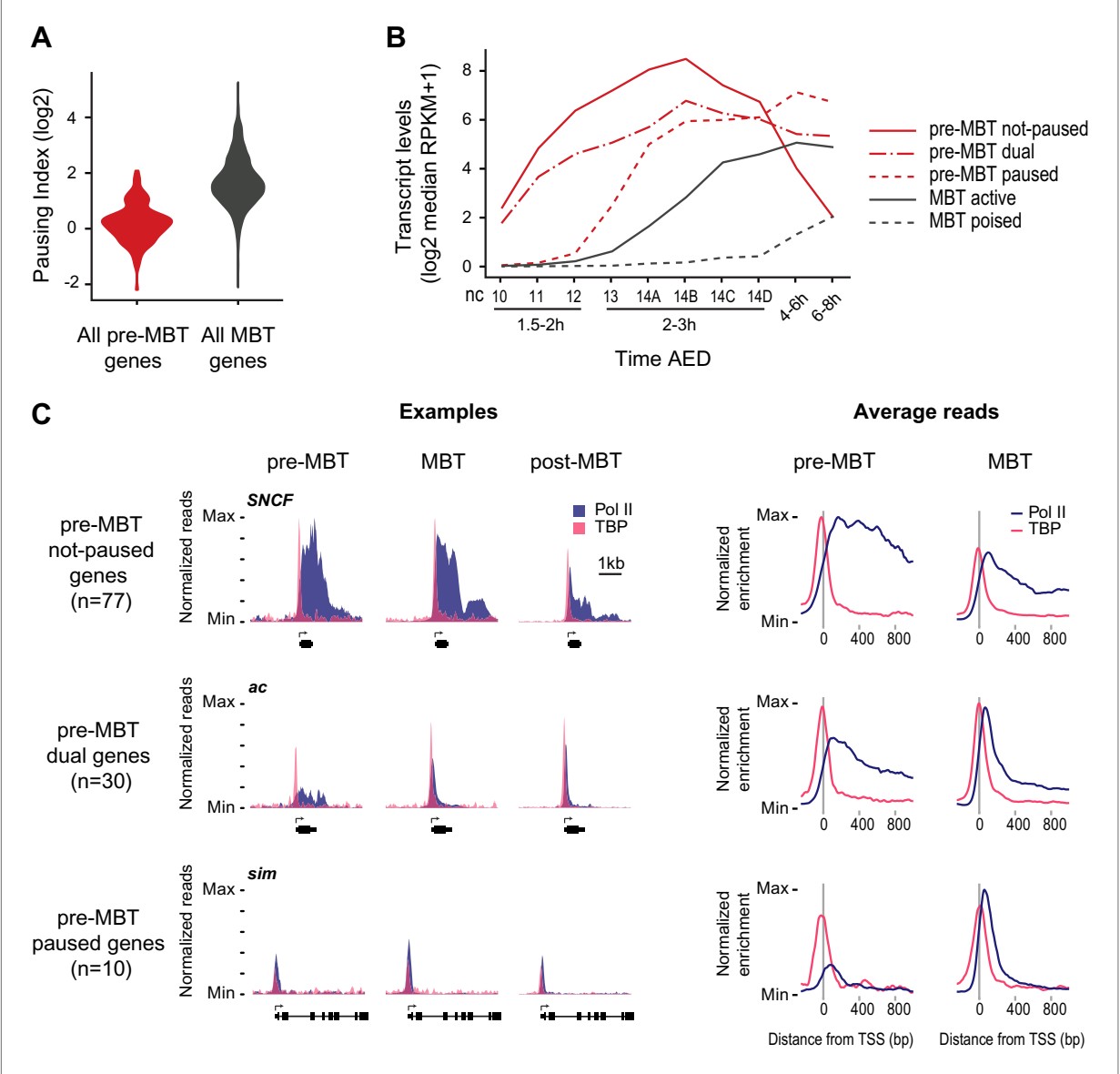

**Figure 2**. Minimal Pol II pausing before the MBT. (**A**) Violin plot of the Pol II pausing index distribution shows that pre-MBT genes (during pre-MBT stages) display less Pol II pausing than MBT genes (during the MBT stage). The width of a violin plot is equivalent to a density curve showing the distribution of values (here pausing indices) within a dataset. (**B**) Median RNA-seq expression data (***Lott et al., 2011***) of the three pre-MBT groups and the two MBT groups shows that paused genes are expressed at lower levels and tend to be induced later. (**C**) Examples and average enrichment of Pol II occupancy (blue) and TBP occupancy (pink) for the three pre-MBT gene groups. Examples are shown as normalized reads while average enrichment is normalized to input. Note that pre-MBT not-paused genes have a non-paused Pol II profile since they do not show elevated Pol II levels at the pause site during pre-MBT and MBT stages. Pre-MBT dual genes switch from an initial non-paused profile during pre-MBT stages to a paused profile during the MBT. Pre-MBT paused genes appear to be paused even during pre-MBT stages. AED = after egg deposition, nc = nuclear cycle.

response elements (PREs) increases over time (***Figure 4B***). Thus, H3K27me3 is likely present in oocytes but may be diluted or erased during replication consistent with a recent study (***Petruk et al., 2012***). Accordingly, it seems unlikely that H3K4me3 or H3K27me27 have a direct role in regulating the zygotic genome activation in *Drosophila*.

We also noticed that pre-MBT genes tend to have particularly low levels of H3K4me3 (***Figure 4C***). Even when pre-MBT genes continue to be transcribed during the MBT, H3K4me3 tends to be low and to accumulate to highest levels at further downstream nucleosomes at pre-MBT genes. This is in stark contrast to the MBT-maternal group, which has a sharp peak of H3K4me3 at the +1 nucleosome. This

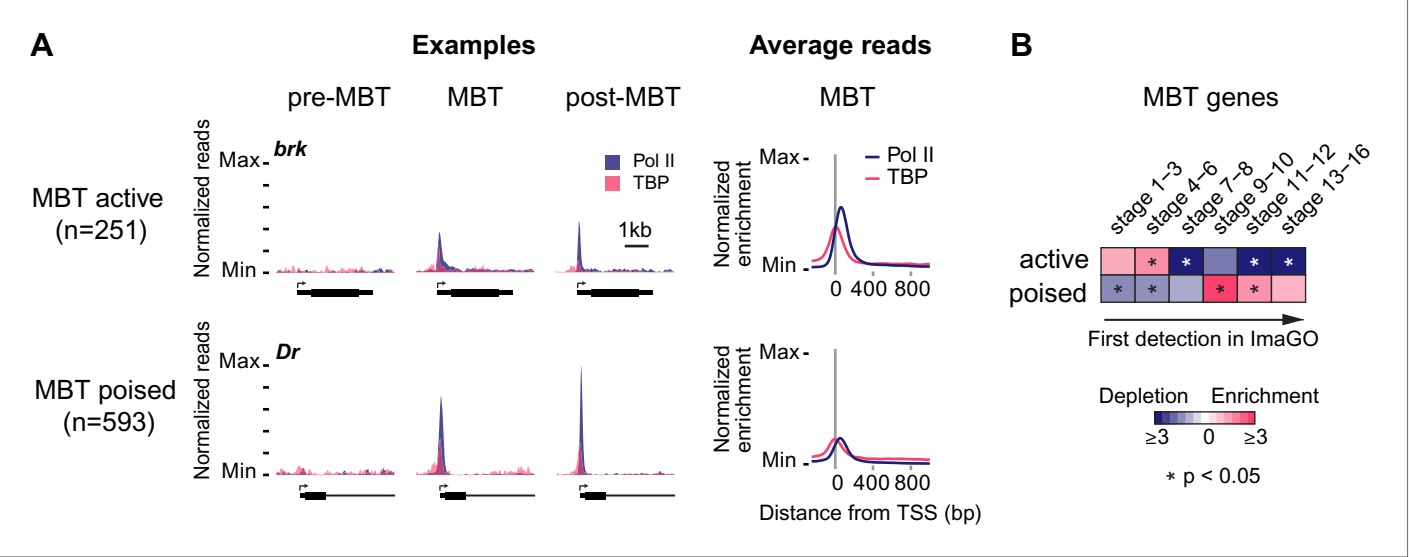

**Figure 3**. Widespread Pol II pausing of MBT genes and poising for later activation. (**A**) Examples and average normalized enrichment (as in **Figure 2C**) of Pol II occupancy (blue) and TBP occupancy (pink) for the MBT-zygotic genes that are significantly transcribed during MBT (MBT active) or not (MBT poised). Both groups show widespread Pol II pausing. (**B**) Analysis of large scale in situ hybridizations (ImaGO database, see 'Materials and methods') confirms the earlier initial expression of MBT active genes (mostly stage 4–6, peri-cellularization) and shows that many MBT poised genes are first transcribed at later stages of embryogenesis (mostly stage 9–10, post-gastrulation).

difference cannot be explained by lower levels of nucleosomes since the nucleosome occupancy based on our Micrococcal Nuclease experiments coupled to deep sequencing (MNase-seq) does not show a dramatic difference between the two gene groups (**Figure 4C**). Since histone modification levels tend to be dependent on the promoter types (**Rach et al., 2011**), we next analyzed whether pre-MBT genes are enriched for specific core promoter elements.

## The different Pol II profiles correlate with distinct promoter types

We analyzed well-studied sequence motifs associated with either focused or dispersed transcription initiation in *Drosophila* (**Table 3**). Focused transcription initiates within a very narrow window and often at a single nucleotide (also called peaked promoters), while dispersed transcription initiates from several weak transcription start sites within a ~50–100 nucleotide region (also called broad promoters) (**Juven-Gershon and Kadonaga, 2010**). In *Drosophila*, dispersed initiation is typically found at broadly expressed housekeeping genes with constitutive promoters. Consistently, we found that MBT-maternal

**Table 2.** Size and intron difference between pre-MBT and MBT zygotic genes

| | Transcript size | | Shortest transcript usage | | Intron content | |
|---|---|---|---|---|---|---|
| **Gene group** | **Gene count** | **Median width (bp)** | **Genes with multiple TSS** | **Genes using shortest transcript (%)** | **Gene count (protein-coding)** | **Genes with no introns (%)** |
| All pre-MBT | 117 | 1228* | 35 | 22 (62.9%)† | 97 | 53 (54.6%)‡ |
| MBT maternal | 3163 | 3322 | 1125 | 308 (27.4%) | 3163 | 228 (7.2%) |
| MBT zygotic | 844 | 6042 | 284 | 87 (30.6%) | 736 | 68 (9.2%) |
| active | 251 | 5422 | 191 | 50 (26.2%) | 251 | 35 (13.9%) |
| poised | 593 | 6771 | 93 | 37 (39.8%) | 485 | 33 (6.8%) |

*Mann–Whitney test for pre-MBT vs MBT zygotic transcript size: $p < 10^{-20}$.
†Fisher test for pre-MBT vs MBT zygotic: $p < 0.0003$.
‡Fisher test for pre-MBT vs MBT zygotic: $p < 10^{-23}$.

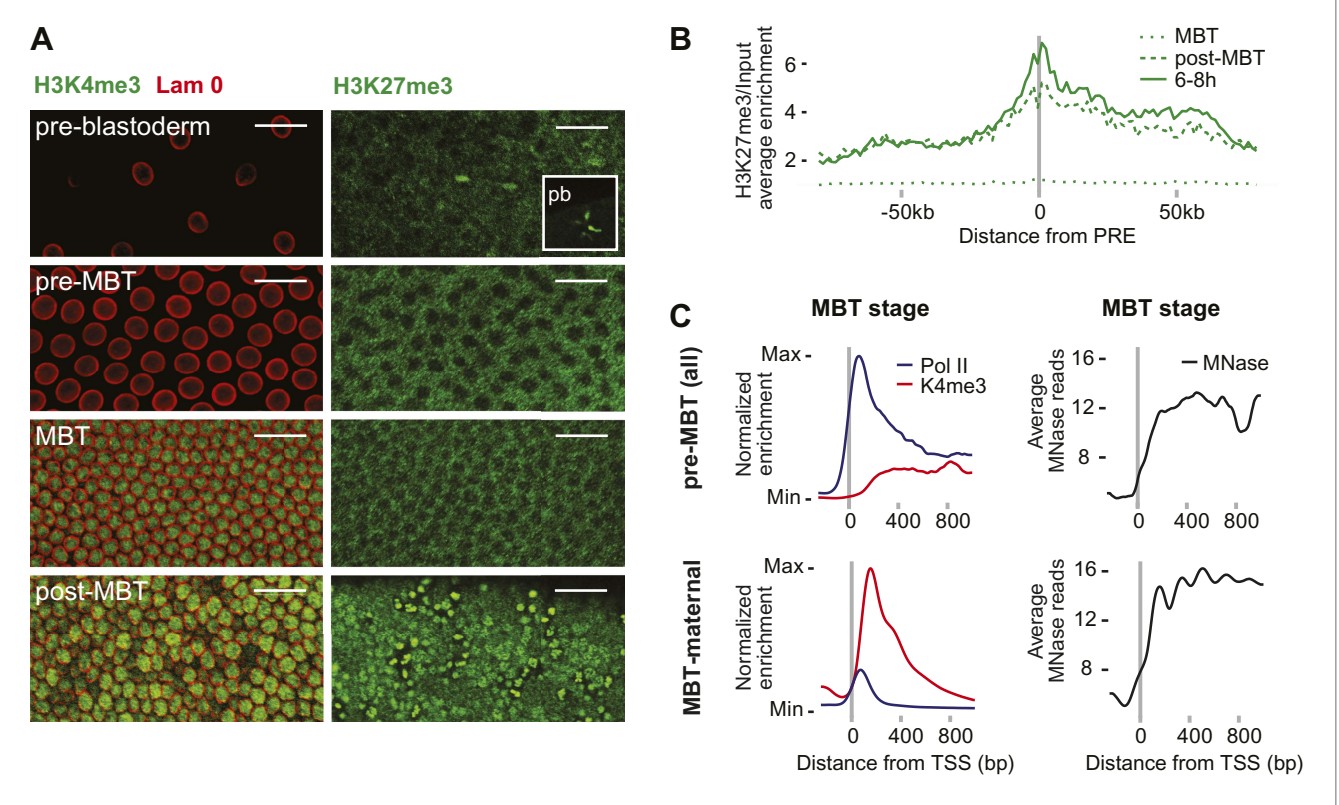

**Figure 4**. Absence of bivalent domains in pre-MBT embryos. (**A**) Lack of detectable H3K4me3 immunostaining in nuclei of embryos before MBT. H3K27me3 is observed in nuclei and polar bodies (pb) of early pre-blastoderm nuclei but not during pre-MBT or MBT stages. H3K27me3 first becomes detectable again in post-MBT embryos (scale = 20 μm). (**B**) ChIP-seq experiments also suggest that H3K27me3 is absent during the MBT but that the levels increase soon after. Shown is the average pattern of H3K27me3 signal over input surrounding 441 previously identified Polycomb response elements (PREs). (**C**) Despite high levels of Pol II occupancy, H3K4me3 average gene enrichments are low at pre-MBT genes (top left). This is in contrast to MBT-maternal genes, which have high H3K4me3 enrichment that peaks ~200 bp downstream of the TSS (bottom left). All enrichments are calculated over input and normalized (see 'Materials and methods'). The overall nucleosome occupancy, as measured by MNase digestion, shows more pronounced nucleosome positioning at MBT maternal genes but the overall nucleosome occupancy is not dramatically different (compare top and bottom panels at the right). Average read counts from a Micrococcal Nuclease (MNase) experiment are shown.

genes are strongly enriched for core promoter elements associated with dispersed initiation: Ohler1, Ohler6, Ohler7 and Dref response element (DRE) (*Figure 5A*). Core promoter elements associated with focused initiation such as Initiator (Inr), downstream promoter element (DPE), Motif Two Element (MTE) and Pause Button (PB) have previously been associated with paused genes (*Hendrix et al., 2008*; *Lee et al., 2008*). As expected, MBT-zygotic genes are highly enriched for these elements, as well as GAGA, which is consistent with reports that GAGA factor (GAF) promotes the recruitment of paused Pol II (*Lee et al., 1992*; *Leibovitch et al., 2002*; *Lee et al., 2008*). We noticed that these genes are not significantly enriched for the TATA box, although they are usually occupied by TBP (*Figure 1C*, and *Figure 3A*).

In contrast, the promoters of pre-MBT genes are significantly enriched for TATA (*Figure 5A*, Supplementary file hosted by Dryad [7.6 Mb]). Interestingly, only pre-MBT genes that initially show the non-paused profile (pre-MBT not-paused, pre-MBT dual) are significantly enriched for Inr and TATA. Furthermore, only pre-MBT genes that are paused at some point (pre-MBT dual, pre-MBT paused) show enrichment for GAGA, Inr and PB (*Figure 5A*). Thus, the presence of specific core promoter elements correlates well with the Pol II occupancy profile across the gene body.

Previous studies have shown that the transcription factor Zelda and its binding motif (known as the TAGteam motif) regulates the onset of zygotic gene expression (*De Renzis et al., 2007*; *Harrison et al., 2011*; *Liang et al., 2008*; *ten Bosch et al., 2006*). Notably, the binding levels of Zelda in the promoter region correlate well with the onset of gene expression, although Zelda is also abundantly found at enhancers (*Harrison et al., 2011*). Consistent with this, we found that Zelda motifs are highly

**Table 3.** *Drosophila* promoter elements analyzed in this study

| Motif name | IUPAC consensus | Directional | Window (bp from TSS) | Transcript count | Reference | Note |
|---|---|---|---|---|---|---|
| DRE | WATCGATW | Yes | −100 to 0 | 2111 | (*Hochheimer et al., 2002*) | |
| Ohler1 | YGGTCACACTR | Yes | −100 to 50 | 609 | (*Ohler et al., 2002*) | Dispersed initiation |
| Ohler6 | YRGTATWTTY | Yes | −150 to 25 | 840 | (*Ohler et al., 2002*) | |
| Ohler7 | CAKCNCTR | Yes | −100 to 50 | 2190 | (*Ohler et al., 2002*) | |
| TATA | STATAWAWR | Yes | −100 to 0 | 1503 | (*Goldberg, 1979*) | |
| Inr | TCAKTY | Yes | −50 to 50 | 5965 | (*Smale and Baltimore, 1989*) | Focused initiation |
| DPE | KCGGTTSK | Yes | 0 to 75 | 537 | (*Burke and Kadonaga, 1996*) | |
| PB | KCGRWCG | Yes | −50 to 100 | 2093 | (*Hendrix et al., 2008*) | |
| MTE | CSARCSSA | Yes | 0 to 30 | 212 | (*Lim et al., 2004*) | |
| GAGA | GAGA | No | −100 to 0 | 9559 | (*Stark et al., 2007*) | Other motifs |
| Zelda | YAGGTAR | No | −2000 to 0 | 9798 | (*Liang et al., 2008; ten Bosch et al., 2006*) | |

enriched in the promoter region of pre-MBT genes (*Figure 5—figure supplement 1*). However, this enrichment is found in all three pre-MBT classes and does not correlate with the Pol II pausing pattern (*Figure 5A*).

To further consolidate the differences between pre-MBT and MBT genes, we performed de novo motif analysis with MEME on the 200 bp centered on the transcription start site (*Figure 5B*). For pre-MBT stage non-paused genes (pre-MBT not-paused, pre-MBT dual), the top two motifs were Zelda and TATA (*Figure 5B*). In contrast, the top two known motifs for the most comparable MBT group (MBT active genes, which are also early-expressed developmental genes), were GAGA and a motif that resembles DPE, MTE and PB (*Figure 5B*). This confirms that pre-MBT and MBT genes differ in their core promoter sequences.

Finally, an analysis of the co-occurrences of core promoter elements similar to previous analyses (*FitzGerald et al., 2006*) also supports our finding (*Figure 5—figure supplement 2*). For example, Zelda, Inr and TATA significantly co-occur among all our Pol II-bound genes (pre-MBT and MBT genes) but not among all annotated genes, suggesting that these motifs preferentially function together during early development.

This suggests a model in which rapid pre-MBT transcription without Pol II pausing is mediated by Zelda bound close to a TATA-enriched promoter (*Figure 6*). In contrast, paused Pol II is typically established through GAF during the MBT at promoters with pausing elements such as DPE, MTE or PB. Thus, there are two principle modes by which zygotic genes are activated but genes may also have elements of both and show a dual behavior.

## Discussion

Our results suggest that a large number of genes first recruit Pol II during the MBT with widespread Pol II pausing, while the small number of genes transcribed before that are not paused. Presumably, these pre-MBT genes are required for very early developmental events such as sex determination (*Erickson and Cline, 1993*; *Barbash and Cline, 1995*) or cellularization (*Schejter and Wieschaus, 1993*; *Pritchard and Schubiger, 1996*), or may be involved in early patterning events that require feedback regulation over time (for example *ftz* [*Edgar et al., 1986*; *Pritchard and Schubiger, 1996*] or *sna* [*De Renzis et al., 2006*; *Reeves et al., 2012*]). Thus, while Pol II pausing is commonly found at developmental genes and may be advantageous for their precise and synchronous expression in response to localized extracellular signals (*Boettiger and Levine, 2009*), a different mode of transcription is used during pre-MBT stages. Due to the short nuclear cycles at this stage, it is likely that transcription is optimized to achieve high levels of transcripts in a very short time period.

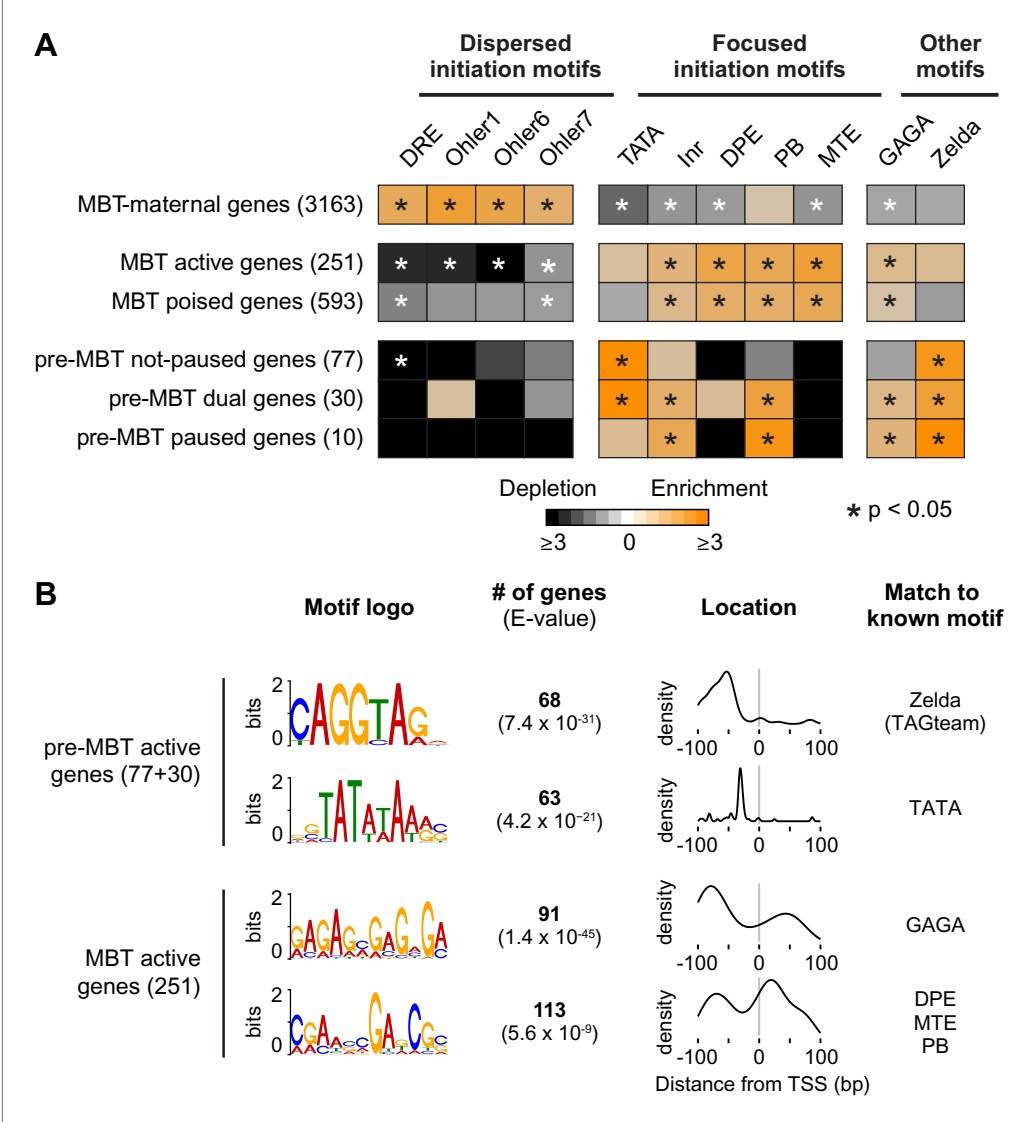

**Figure 5**. Differential usage of core promoter elements during the zygotic genome activation. (**A**) Promoter analysis of all previously identified gene groups. Shown is the enrichment of known core promoter elements found in promoters with dispersed initiation or focused initiation, as well as in the binding motifs for GAGA factor (GAF) and Zelda. Only occurrences close to the known location of the motif relative to the TSS were scored (see **Table 3**). The star indicates significant enrichment (orange) or depletion (black). Note that the three pre-MBT groups with different Pol II pausing patterns are enriched for distinct core promoter elements. (**B**) The top two known motifs identified by de novo motif analysis for active pre-MBT genes and active MBT genes. The analysis was performed with MEME on the 200 bp long region centered on the TSS. The number of occurrences, p-value and the density distributions relative to the TSS of the identified motifs are shown on the right. Note that all motifs are found with the highest frequency at the expected location but that the DPE/MTE/PB is less specific and more frequently found at positions where it is unlikely to be functional.

The following figure supplements are available for figure 5:

**Figure supplement 1**. High frequency of Zelda motifs in the promoter region of pre-MBT genes.

**Figure supplement 2**. Analysis of the co-occurrence of promoter elements among all genes and early genes.

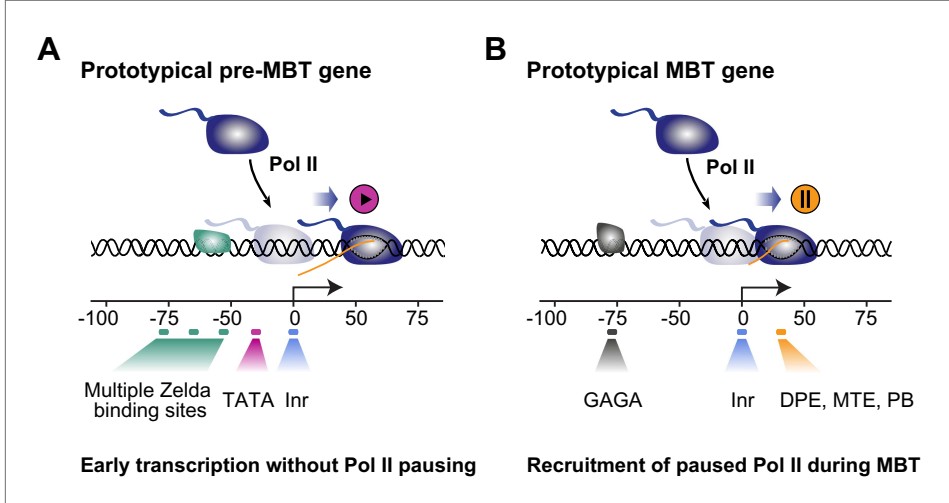

**Figure 6**. Proposed model for the two main modes of Pol II recruitment and elongation behavior during the zygotic genome activation. Before the midblastula transition, when the cell cycle is fast, efficient transcription occurs through a TATA promoter with multiple Zelda sites upstream. This combination leads to transcription without pausing, presumably due to fast re-initiation. During the midblastula transition, Pol II is recruited de novo to many genes and pausing is established with the help of GAGA factor and core promoter elements associated with pausing such as DPE, MTE and PB. Note that genes can have core promoter elements of both modes (e.g., TATA and PB), leading them to switch from a non-pausing behavior to a pausing behavior during the midblastula transition. It is likely that transcription factors in addition to Zelda and GAF also influence the Pol II behavior at genes.

The fact that we find the TATA box enriched among pre-MBT genes is consistent with the known properties of TATA-containing promoters. TATA is a strong core promoter element that efficiently supports transcription in vitro (*Aso et al., 1994*), mediates efficient re-initiation in vitro (*Yean and Gralla, 1997*, *1999*), and its presence in vivo correlates with 'bursts' of transcription that produce many transcripts within a short time (*Zenklusen et al., 2008*). Furthermore, it has been shown that TATA promotes pTEFb activity, leading to more efficient elongation rates in vitro and in vivo (*Amir-Zilberstein et al., 2007*; *Montanuy et al., 2008*).

This suggests that TATA-enriched promoters and paused promoters have different transcription dynamics and serve different purposes during development. While this difference is particularly evident during the zygotic genome activation as reported here, we propose that this difference is general and also applies to later development. For example, we have analyzed promoters during muscle development and find that many genes are induced during late stages of embryogenesis without prior Pol II pausing and that these genes are also enriched in TATA (*Gaertner et al., 2012*). Consistent with this, statistical analyses suggest that the combination of Inr and TATA represents a separate class of promoters that is often found among genes expressed in adult tissues (*FitzGerald et al., 2006*; *Lenhard et al., 2012*). Since the properties of TATA are not specific to *Drosophila*, it is likely that differences among promoter types and their propensity for Pol II pausing are conserved across animals.

## Materials and methods

### Embryo collection and immunostainings

Wild-type embryos (Oregon R) were collected from six population cages (28 × 17 × 17 cm, with 10,000–12,000 flies each, maintained in fly incubators at 25°C and 60% humidity) on 15 cm apple juice plates with yeast paste after pre-clearing. The collection windows were 0–4 hr after egg deposition (AED) for immunostainings, and 1–2 hr, 2–3 hr, 2–4 hr, 6–8 hr AED for ChIP. Embryos were dechorionated with bleach and cross-linked with 1.8% formaldehyde in 2.5 ml Hepes buffer and 7.5 ml heptane, while vortexing at medium speed for 15 min. Embryos were devitellinized in methanol/heptane and kept at −20°C in methanol for up to 3 months until needed. Immunostainings were performed by standard methods with Alexafluor Dyes and the embryos were analyzed by confocal imaging (Zeiss LSM-510-VIS, Carl Zeiss Microscopy, LLC, Thornwood, NY).

## Antibodies

The following antibodies were used for immunostainings: mouse monoclonal antibody CTD4H8 against Pol II CTD (05-623; EMD-Millipore, Billerica, MA), mouse monoclonal antibody H14 against Ser5-phosophorylated Pol II (MMS-134R; Covance, Princeton, NJ), rabbit polyclonal antibodies against H3K4me3 (9751S; Cell Signaling Technology, Danvers, MA), rabbit polyclonal antibodies against H3K27me3 (39155; Active Motif, Carlsbad, CA), rabbit polyclonal antibodies against dTBP (a kind gift from J Kadonaga), and mouse monoclonal antibody against lamin 0 (ADL101 developed by P Fisher, obtained from the Developmental Studies Hybridoma Bank). Embryos were visualized by confocal imaging (LSM-510-VIS).

The following antibodies were used for ChIP: antibodies against dTBP (a kind gift from J Kadonaga), H3K4me3 (9751S; Cell Signaling Technology), H3K27me3 (39155; Active Motif), mouse monoclonal Pol II antibody 8WG16 (MMS-126R; Covance) and rabbit polyclonal antibodies against H3ac (07-360; Millipore).

## Embryo sorting

For embryo sorting, wild-type embryos collected within 1–2 hr (pre-MBT) and 2–3 hr (MBT) were fixed in 1.8% formaldehyde and stained with DAPI. Embryos were sorted in PBT on ice under an inverted contrasting microscope (Leica DMIL, Buffalo Grove, IL). All embryos were screened once for morphology with a DIC filter, and twice for DNA content under UV light (*Figure 1—figure supplement 1*). Out-of-stage embryos were removed with a 10 µl pipette tip connected to a Cell Tram Vario (920002111; Eppendorf, Hauppauge, NY). After practice, 200 µl embryos (~5000 embryos) could be screened in 1 day.

## ChIP-sequencing

ChIPs were performed with whole cell extracts (WCE) from embryos as previously described (*Zeitlinger et al., 2007*) using 200 µl embryos for pre-MBT embryos and 50 µl for older embryos. Libraries from the immunoprecipitated DNA and WCE DNA were prepared using the Paired-End DNA Sample Preparation Kit (PE-102-1001; Illumina, San Diego, CA) but with a modified protocol. To remove adapter dimers, biotin-labeled dATP (19524-016; Invitrogen, Grand Island, NY) was added in the A-tailing reaction after end-repair. After ligation to the PE adaptor, the samples were incubated with streptavidin beads in 500 µl binding buffer (650-01; Invitrogen) at room temperature for 15 min. DNA bound to the beads was then washed twice with 800 µl binding buffer with 0.05% Tween 20, twice with NEB buffer 2 (New England Biolabs, Ipswich, MA) and resuspended in 31 µl NEB buffer 2. The PCR reaction was then performed according to the Illumina protocol.

## Nucleosome mapping

50 µl sorted 2–3 hr embryos were homogenized as previously described (*Zeitlinger et al., 2007*) and digested based on a previously published protocol (*Mavrich et al., 2008*). Briefly, homogenized chromatin in NPS buffer was digested with an MNase (LS004798; Worthington, Lakewood, NJ) gradient of 20 U, 10 U, 5 U, 5/2 U, 5/4 U, 5/8 U, 5/16 U, to 5/32 U, and a negative control for 30 min at 37°C. Mono-nucleosome size DNA was extracted from the lane with two clear bands in a 1.7% agarose gel, and prepared for paired-end sequencing.

## Alignment of ChIP-seq and MNase-seq data

All sequencing reads were aligned to the UCSC *Drosophila melanogaster* dm3 genome with Bowtie v0.12.8 (*Langmead et al., 2009*) using the following parameters:

-k 1 –m 1 –l 40 –n 2 –best –strata

The MBT MNase-seq library was paired-end sequenced and alignment was performed with an allowable insert size of 47 bp to 297 bp. After alignment, single-end reads were extended to the estimated insert size of the library as determined by a Bioanalyzer. To identify alignment and amplification artifacts, custom R scripts were used to analyze the aligned reads of all single-end libraries with more than 10 duplicates (defined as having the same chromosome, start and strand values). These 'stacks' of identical reads were removed unless a corresponding number of reads were present on the opposite strand approximately one fragment length away in the 3′ direction. For all libraries, genome-wide coverage was calculated by assigning an integer score to each genomic coordinate representing the number of extended reads that overlapped that location.

## Analysis of RNA-seq expression data

To obtain gene expression measurements at different nuclear cycles, we downloaded single-embryo RNA-seq datasets from http://eisenlab.org/dosage/ (*Lott et al., 2011*). One female and one male replicate were downloaded for nuclear cycles 10, 11, 12, 13 and 14 (A-D). The male and female datasets were combined for each nuclear cycle. In addition, we downloaded 4–6 hr and 6–8 hr staged whole-embryo RNA-seq datasets (*Graveley et al., 2011*). We processed all samples using TopHat v2.0.4 (*Trapnell and Salzberg, 2009*) by aligning against the FlyBase r5.47 genome and its corresponding gene annotations using the following parameters:

Single-embryo samples (40 bp reads):

-G fb-r5.47.gtf –I 5000 –segment-length 20 fb547_genome

4–6 hr and 6–8 hr embryo samples (75 bp reads):

-G fb-r5.47.gtf–I 5000 –segment-length 37 fb547_genome

Next, we used the cuffdiff tool from Cufflinks v2.0.2 (*Trapnell et al., 2010*) to obtain gene expression values (RPKMs) for all samples using the following non-default parameters:

-u –b fb547_genome.fa fb-r5.47.gtf

## Calculating Pol II enrichments and stalling indexes

For the four Pol II replicates in the pre-MBT embryo and the three Pol II replicates in the MBT embryo, enrichment ratios were calculated for the TSS region (first 200 bp of the transcript), a region immediately downstream of the TSS (+201 to +400 bp), and the transcription unit (TU) region (+401 to the end of the transcript) of each unique FlyBase r5.47 transcript. For transcripts less than 600 bp in length, the TU region was defined as the entire transcript. Total signal for each region was found for each Pol II and WCE sample. Enrichment in each region was calculated after normalizing for both fragment length and total read count:

$$\text{Enrichment} = (\text{IP signal}/[\text{IP read count} \times \text{IP fragment length}])/$$
$$(\text{WCE signal}/[\text{WCE read count} \times \text{WCE fragment length}])$$

The stalling index for each gene was defined as: $\log2\ \text{Pol II}_{TSS} - \log2\ \text{Pol II}_{Downstream\ TSS}$ after flooring both Pol II enrichment values at 1 (background). Stalling indexes for all replicates were averaged.

## Identification and classification of pre-MBT genes

To identify genes bound by Pol II in the pre-MBT embryo, we first identified all transcripts with Pol II$_{TSS}$ enrichment twofold above WCE in all four replicates. To ensure that these enrichments were due to high Pol II signal, we also required the Pol II signal portion of the enrichment calculation (the numerator in the above equation) to be in the 99th percentile of all transcripts in all four replicates.

Manual inspection of some of these transcripts showed that the Pol II signal originated from a different gene's TSS (see examples in *Figure 1—figure supplement 4*). To eliminate these false positives, we used MACS to identify peaks in our best pre-MBT TBP sample and manually examined all Pol II-enriched pre-MBT transcripts that did not have a detected TBP peak within 500 bp of the TSS. We used the default parameters of MACS v2.0.10.20120703 (*Zhang et al., 2008*), specifying only the preset alignable genome size for *Drosophila melanogaster* using the '-g dm' argument. This identified 12 transcripts in which the Pol II signal did not appear to originate from the TSS. These transcripts were removed from our pre-MBT list and are marked as 'rejected pre-MBT genes' in (Supplementary file hosted by Dryad [7.6 Mb]).

We next checked for possible pre-MBT genes with missing or mis-annotated transcription start sites. To do this, we used MACS to call peaks on all four of our pre-MBT Pol II samples using the same default parameters as described above. We then identified all regions that were called as peaks in at least two of the four replicates. These regions were assigned to the nearest gene within 5 kb and all regions assigned to a gene not already considered a pre-MBT gene were manually examined. This revealed ten possible additional pre-MBT genes where the Pol II signal originated from an un-annotated transcription start site. As all of these genes also had at least some TBP signal upstream of the Pol II signal, we defined custom transcript entries for these genes by setting the transcript start site to 19 bp downstream of the location of the maximum TBP signal. To ensure these custom transcripts met our existing enrichment criteria, we performed the same calculations as described above in the Calculating Pol II enrichments section. All ten of the custom transcripts were sufficiently enriched in Pol II and were added to our pre-MBT gene list.

We classified the 117 pre-MBT genes into three groups. First, the 'paused' group was defined as those pre-MBT genes having a mean (among all four replicates) Pol II$_{TU}$ ratio less than 1. The 'dual' group was defined as any pre-MBT gene not in the paused group that had Pol II$_{TSS}$ enrichment in the top 20% of all genes in 6–8 hr Mef2-positive muscle cells (*Gaertner et al., 2012*). The remaining pre-MBT genes were classified as the 'not paused' group.

## Identification and classification of MBT genes

To identify genes bound by Pol II in the MBT embryo, we selected all transcripts with Pol II$_{TSS}$ enrichment at least twofold above WCE in all three replicates. If multiple transcripts for the same gene met these criteria, we selected the one with the highest Pol II$_{TSS}$ signal (breaking ties using the mean Pol II$_{TU}$ enrichment).

MBT genes were classified into three groups using gene expression values calculated from previously published single-embryo RNA-seq experiments (see 'Analysis of RNA-seq expression data' section). We classified as 'maternal' all MBT genes with an RPKM of at least 1 in nuclear cycle 10. We classified as 'MBT active' all non-maternal MBT genes with an RPKM of at least 5 in nuclear cycle 14D. The remaining MBT genes were classified as 'MBT paused'.

## Conservation analysis

PhastCons scores (*Siepel et al., 2005*) from the alignment of 14 insect species' genome assemblies to the *Drosophila melanogaster* genome were downloaded from http://hgdownload-test.cse.ucsc.edu/goldenPath/dm3/phastCons15way/. For each mRNA transcript in Flybase r5.47, the mean phastCons score along the transcript's length was used as a relative measure of conservation. These transcripts were then organized into groups as described in the 'Identification and classification of pre-MBT genes' and 'Identification and classification of MBT genes' sections.

## Normalization of reads and enrichment values

For *Figure 1C*, enrichment values were first calculated in a 100 bp sliding window across all samples. Replicates were combined by taking the minimum enrichment value at each base. Samples were then independently normalized by defining 'minimum' as an enrichment value of 1 (background) and 'maximum' as the 99th percentile enrichment value encountered among all displayed bases. As there was no significant ChIP enrichment in the pre-MBT H3K4me3 sample, it was normalized to the maximum enrichment value of the MBT H3K4me3 sample to avoid amplifying noise.

For *Figure 2*, *Figure 3C* and *Figure 1—figure supplement 3*, both read counts and enrichment values for each sample were independently scaled by dividing the values at each base by the maximum value encountered among the displayed genes or gene groups across stages after normalizing for both read count and fragment size differences.

## Metapeak analysis for H3K27me3 ChIP-Seq

Using supplemental Table 17 from Schuettengruber et al. (*Schuettengruber et al., 2009*), the 441 regions were selected based on Ph ChIP-chip enrichment (p<0.0001). The regions were then aligned at their midpoints and extended by 80 kb in both directions. Average region graphs were constructed showing the average enrichment value at each base for three H3K27me3 samples. The enrichment for each sample was defined by dividing read-count normalized IP signal by the read-count normalized WCE control signal.

## Promoter element annotations

Sequences surrounding all FlyBase r5.47 transcription start sites (plus our ten additional custom pre-MBT transcripts) were scanned for the core promoter elements listed in *Table 3*. A core promoter element was scored as present if found with no mismatch within a specified window relative to the transcription start site. For Zelda, we also counted the number of motifs found in each transcript's window.

## Promoter element enrichments

For each group of transcripts analyzed for promoter element composition, an enrichment and p-value were calculated for each promoter element. Enrichment was calculated as follows, where G is the group of transcripts tested and PE is a particular promoter element:

$$\text{Observed} = \text{(Number of transcripts in G with element PE)} / \text{(Number of transcripts in G)}$$

$$\text{Expected} = \text{(Number of transcripts in the genome with element PE)} / \text{(Number of transcripts in the genome)}$$

$$\text{Enrichment} = \text{Observed/Expected}$$

For enrichment values less than one, the negative reciprocal of the enrichment value was used (indicating depletion instead of enrichment). To calculate a p-value for the observed frequency of each promoter element in each group of transcripts, a Fisher test was performed. Enrichments and depletions with a $p<0.05$ (after correcting for multiple testing with the Benjamini and Hochberg method) were deemed significant.

For Zelda, we calculated enrichment and p-values via random sampling. Enrichment values for each group of transcripts were calculated by dividing the number of Zelda sites per transcript in each group by the average number of Zelda sites per transcript in the genome. To calculate p-values for each group of transcripts, we randomly selected an equal number of transcripts from the entire genome 10,000 times and calculated the enrichment value for each random sample. The p-value was then calculated as the portion of random samples with higher Zelda enrichment than the transcript group.

## Promoter element co-occurrence analysis

The enrichments (Observed/Expected) and p-values (Fisher test) for the co-occurrence of promoter elements were calculated for all pairs of core promoter elements in three sets of genes. For the 'FitzGerald' set of genes, we extracted the gene and overlap counts for a subset of motifs listed in *Table 1* of *FitzGerald et al., 2006*. For the 'all promoters' set, we included all unique promoters in FlyBase r5.47 as well as our additional ten custom transcripts. For the 'pre-MBT and MBT' set, we included only the promoters of our pre-MBT and MBT gene groups. The order of motifs shown in *Figure 5—figure supplement 2* was determined by hierarchically clustering the enrichment values with the R v3.0.1 *hclust* function using Euclidean distance. For the Zelda motif, which can occur multiple times in a single promoter region, the presence of at least one motif within 2 kb upstream was scored.

## De novo motif discovery

Fasta sequence files were generated for each of the classified Pre-MBT and MBT groups, based on regions ±100 bp surrounding the FlyBase r5.47 transcription start sites (plus our ten additional custom pre-MBT transcripts). Using MEME v4.8.1 (*Bailey et al., 2009*) the fasta files were processed using the following parameters:

    -mod zoops -dna -nmotifs 50 -revcomp -maxw 12 -maxsize 5000000 -oc meme/

The resulting motifs were then compared against the TRANSFAC 2011.4 database and the *Table 3* listed above using the TOMTOM tool also from the MEME suite.

## ImaGO earliest annotated expression analysis

To calculate enrichment of gene groups for their first expression in specific tissues, we downloaded the Berkeley *Drosophila* Genome Project in situ expression database from http://insitu.fruitfly.org/ (*Tomancak et al., 2002*, *2007*). We removed the 'maternal' and 'no staining' annotation entries and then removed all but the first (in stage order) annotation of each gene. For each gene group analyzed, we calculated enrichments and p-values using the same method as the promoter element enrichment analysis described above.

## Analysis of gene expression over time

For all gene groups plotted in *Figure 2B* and *Figure 1—figure supplement 4*, we first removed any genes with evidence of maternally deposited mRNA using the following criteria:

- RPKM expression > 16 in nuclear cycle 10, or
- Maternal expression at least twofold above zygotic expression in nuclear cycle 10 [F10 sample in Dataset S1 of S. E. Lott et al. (*Lott et al., 2011*)]

## Open access of data and analysis

All ChIP-seq and MNase-seq data have been deposited with the NCBI Gene Expression Omnibus under accession number GSE41703. In addition, we have replicated our analysis environment (including software tools, analysis source code, and raw data) in a Linux virtual machine hosted by Amazon Web Services. Instructions for accessing the virtual machine can be found at http://research.stowers. org/zeitlingerlab/data. The analysis code is available on GitHub at https://github.com/zeitlingerlab/ chen_elife_2013.

## Supplementary spreadsheet

A spreadsheet summarizing our classification and analysis of pre-MBT and MBT gene groups is available via Dryad digital repository. The first sheet gives an explanation for all column headings. The second sheet lists all data for all our annotated genes, including our ten custom transcripts. It includes the classifications into pre-MBT and MBT gene groups, the Pol II ChIP-seq enrichment values at the transcription start site (TSS) and transcription unit (TU) for all replicates, phastCon conservation scores, and the presence or absence of all core promoter motifs analyzed in this study, as well as the presence of the TATA element identified by de novo motif analysis.

## Acknowledgements

We thank A Shum for help with the gene conservation analysis, J Kadonaga for TBP antibodies, and J Conaway and R Krumlauf for comments on the manuscript. KC is a PhD student registered with the Open University, UK.

## Additional information

### Funding

| Funder | Grant reference number | Author |
|---|---|---|
| National Institutes of Health New Innovator Award | 1DP2 OD004561-01 | Julia Zeitlinger |
| Pew Charitable Trust | | Julia Zeitlinger |

The funders had no role in study design, data collection and interpretation, or the decision to submit the work for publication.

### Author contributions

KC, Conceived and designed the project, developed the sorting technique and ChIP protocol for early embryos and performed experiments, analyzed and interpreted the data, wrote the manuscript; JJ, Designed, organized, and performed data analysis, including quality control, normalization, gene classifications, and core promoter element analysis, wrote the manuscript; WS, Performed embryo sorting and ChIP-seq experiments, provided input to the manuscript; SM, Performed data analysis, including the de novo motif analysis and its preparation into a figure; CS, Designed and organized embryo collections, provided input to the manuscript; JZ, Conceived and designed the project, analyzed and interpreted the data, wrote the manuscript

## Additional files

### Major dataset

The following datasets were generated:

| Author(s) | Year | Dataset title | Dataset ID and/or URL | Database, license, and accessibility information |
|---|---|---|---|---|
| Chen K, Johnston J, Shao W, Meier S, Staber C, Zeitlinger J | 2013 | A global change in RNA Polymerase II pausing during the *Drosophila* midblastula transition | GSE41703; http://www. ncbi.nlm.nih.gov/geo/ query/acc.cgi?acc= GSE41703 | Publicly available at GEO (http://www.ncbi. nlm.nih.gov/geo/) |

| Chen K, Johnston J, Shao W, Meier S, Staber C, Zeitlinger J | 2013 | Data from: A global change in RNA Polymerase II pausing during the *Drosophila* midblastula transition | 10.5061/dryad.rv624; http://dx.doi.org/ 10.5061/dryad.rv624 | The spreadsheet summarizing our classification and analysis of pre-MBT and MBT gene groups is publicly available at Dryad (http:// datadryad.org/) |

The following previously published datasets were used:

| Author(s) | Year | Dataset title | Dataset ID and/or URL | Database, license, and accessibility information |
| --- | --- | --- | --- | --- |
| Lott SE, Eisen MB | 2011 | Non-canonical compensation of zygotic X transcription in *Drosophila melanogaster* development revealed through single embryo RNA-Seq | GSE25180; http://www. ncbi.nlm.nih.gov/geo/ query/acc.cgi?acc= GSE25180 | Publicly available at GEO (http://www.ncbi. nlm.nih.gov/geo/) |
| modENCODE Project | 2011 | Developmental Time Course poly(A)+ RNA Profiling in *D. melanogaster* | SRP001065; http:// trace.ncbi.nlm.nih.gov/ Traces/sra/?study= SRP001065 | Publicly available at the Sequence Read Archive (http://www. ncbi.nlm.nih.gov/sra). |

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
