## [Decision Letter]

Thank you for sending your work entitled “A global change in RNA Polymerase II pausing during the *Drosophila* midblastula transition” for consideration at *eLife*. Your article has been favorably evaluated by a Senior editor, a Reviewing editor, and 2 reviewers. The Reviewing editor and reviewers discussed their comments before we reached this decision, and the Reviewing editor has assembled the following comments to help you prepare a revised submission.

We are delighted to tell you that the review of your manuscript is now completed and the reviewers really liked your study. We suggest a few small changes that we are sure you can do in short time. Please below see the combined reviews.

This study constitutes an important systematic contribution to the analysis of the expression profile and chromatin state of the *Drosophila* embryo, before, during, and after the midblastula transition. Pre-MBT embryos were carefully hand-sorted and analyzed by ChIP-seq for Pol II, TBP, and histone modifications (Figure 1). Strong occupancy of TBP and Pol II as well as low levels of H3K4me3 were observed on the pre-MBT genes in the pre-MBT embryos (Figure 1). By using a pausing index based on the distribution of Pol II, it was found that most (77 to 107 out of 117) pre-MBT genes exhibit little transcriptional pausing (Figure 2). In contrast, a strong extent of pausing was observed with genes that are activated during the MBT (Figure 3). H3K4me3 and H3K27me3 were essentially absent from the nuclei of pre-MBT embryos (Figure 4). Analysis of core promoter sequences revealed that the TATA box is significantly enriched in the promoters of pre-MBT genes (Figure 5). These findings lead to the model shown in Figure 6. In essence, zygotic transcription from pre-MBT genes appears to occur typically via the TATA box by a mechanism that does not involve transcriptional pausing.

The findings suggest that there is a specific, perhaps streamlined, mechanism of transcription involving the TATA box that is used to achieve the rapid synthesis of transcripts that is needed in the pre-MBT embryo. This is a notable and interesting finding.

Prior to publication we would like to see the following changes made to the manuscript. These changes are largely aimed at making the manuscript more easily accessible to the readership.

1) The authors need to provide some statistics on reproducibility among the biological replicates used.

2) While the authors will release a GEO accession, having supplements at the journal site is a good idea. Supplementary file 1 is critical (the most important for anyone really wanting to study the primary results), but is lacking in ways that will make it difficult to use. There should be another sheet in the workbook, or a separate readme file that has all the def lines and nomenclature explained. For example, I have no idea what “Custom_slam” means in a column where all the rest of the cells appear to have FBtr#s. I can guess what most of the columns labels mean, but you should not make the reader do this. Additionally, many of the important data fields in Supplementary file 1 are conclusions rather than data (e.g., “yes”, “never-paused”, etc). These should be replaced with numbers used for the classification system as outlined in the methods. The classification scheme itself is somewhat ad hoc and in a different study, by a different lab, so it is almost certain that some of the same genes will be classified differently. These types of data handling issues have muddled many areas of genomics and can be ameliorated by giving others data, allowing them to apply a different classifier.

3) Write something about the nucleosome positioning in Figure 4.

4) The core promoter enrichments are clear in Figure 5, but are there any dependencies of the type mentioned in the Discussion? For example, in the case of preMBT dual genes, do they have an Inr and a PB element? One or the other?

5) In Figure 5, the locations of the putative DPE, MTE and PB sites are distributed throughout the promoter region. It is known, however, that these sequences function only in a specific downstream location. Therefore, DPE, MTE, and PB-like sequences that are not located in their characteristic downstream location should probably be disregarded.

6) There are a confusing number of core promoter and MBT classification names. This is certainly not all the fault of the authors, but it would help the reader if the names from the traditional literature and the computational studies were both used or listed somewhere to help match Ohler and FitzGerald, for example. This could be added to the “promoter element table”. The authors should not send the reader to another paper for a term that they could easily define in the manuscript (dispersed and focused are never described). The long descriptors (“pre-MBT not-paused genes”) could be replaced in the figures with data in the heat maps. Almost anything that would make the high-dimensional naming of promoters easier to follow would be greatly appreciated.

---

## [Author Response]

*1) The authors need to provide some statistics on reproducibility among the biological replicates used*.

We agree that additional information about the reproducibility of our ChIP-seq experiments would be useful. In addition to the existing scatterplots comparing two replicates each of pre-MBT and MBT Pol II_TSS_ enrichments in Figure 1—figure supplement 1, we have calculated Pearson correlations for the four pre-MBT Pol II replicates and the three MBT Pol II replicates. The correlations are all very high, ranging from 0.78 to 0.92 for the pre-MBT replicates and from 0.96 to 0.98 for the MBT replicates, in comparison to 0.16 to 0.33 when these replicates are correlated with input (WCE). They are now shown in panel B of Figure 1—figure supplement 1.

*2) While the authors will release a GEO accession, having supplements at the journal site is a good idea. Supplementary file 1 is critical (the most important for anyone really wanting to study the primary results), but is lacking in ways that will make it difficult to use. There should be another sheet in the workbook, or a separate readme file that has all the def lines and nomenclature explained. For example, I have no idea what “Custom_slam” means in a column where all the rest of the cells appear to have FBtr#s. I can guess what most of the columns labels mean, but you should not make the reader do this. Additionally, many of the important data fields in Supplementary file 1 are conclusions rather than data (e.g., “yes”, “never-paused”, etc). These should be replaced with numbers used for the classification system as outlined in the methods. The classification scheme itself is somewhat ad hoc and in a different study, by a different lab, so it is almost certain that some of the same genes will be classified differently. These types of data handling issues have muddled many areas of genomics and can be ameliorated by giving others data, allowing them to apply a different classifier*.

We apologize for the lack of full descriptions in the spreadsheet and we have substantially revised it to improve its clarity and usefulness. As suggested, we have added an additional sheet containing complete descriptions of each column. In addition, we have added all data columns used in our gene classification schemes.

*3) Write something about the nucleosome positioning in*
Figure 4.

We now say in the legend of Figure 4: “The overall nucleosome occupancy, as measured by MNase digestion, shows more pronounced nucleosome positioning at MBT maternal genes but the overall nucleosome occupancy is not dramatically different (compare top and bottom panels at the right).”

We did not emphasize this result in the main text because the strong positioning of nucleosomes at promoters with dispersed initiation has been observed before ([44] and [17]) and the number of pre-MBT genes is relatively small.

*4) The core promoter enrichments are clear in*
Figure 5*, but are there any dependencies of the type mentioned in the Discussion? For example, in the case of preMBT dual genes, do they have an Inr and a PB element? One or the other*?

Thank you for the suggestion. We have now added a figure on the co-occurrence of promoter elements (Figure 5—figure supplement 2). Our results agree very well with previous results (15) and show that the association between Zelda, TATA, and Inr is specific for early development.

We have added this analysis and mention this result in the main text: “Finally, an analysis of the co-occurrences of core promoter elements similar to previous analyses (15) also supports our finding (Figure 5—figure supplement 2). For example, Zelda, Inr, and TATA significantly co-occur among all our Pol II-bound genes (pre-MBT and MBT genes) but not among all annotated genes, suggesting that these motifs preferentially function together during early development.”

The figure also shows that Inr and PB significantly co-occur. Note, however, that most co-occurrences are only significant when analyzing a large data set and not in selected pre-MBT or MBT gene sets. Thus, it is likely that promoters do not generally require a pair of motifs with a perfect match to the consensus sequence (i.e., a high affinity binding site). We interpret a co-occurrence of two motifs as evidence that they are found in the same promoter type and can function together.

*5) In*
Figure 5*, the locations of the putative DPE, MTE and PB sites are distributed throughout the promoter region. It is known, however, that these sequences function only in a specific downstream location. Therefore, DPE, MTE and PB-like sequences that are not located in their characteristic downstream location should probably be disregarded*.

We agree that instances of DPE, MTE and PB motifs found upstream of the transcription start site in the MBT active gene group are unlikely to be functional. That’s why we selected a specific window relative to the transcription start site when we scanned for known motif sequences in Figure 5. These windows are listed in the promoter element table in the methods and were based on previous experimental evidence and our own computational analyses. In contrast, Figure 5 shows a de novo motif analysis and the distribution for each recovered known motif. This shows that the most frequent location of the identified motifs agrees with previous studies. Although the recovered DPE/MTE/PB motif is not as specific in its location, we felt that it would be wrong to manipulate the original results. Rather, our results suggest that the recovered DPE/MTE/PB motif is more frequently found by chance (outside its location where it is functional) as compared to the other motifs.

We apologize for the confusion this has caused and added to the legend of Figure 5. “Only occurrences close to the known location of the motif relative to the TSS were scored (see promoter element table in the Materials and methods).” Furthermore, we added to the legend of Figure 5: “Note that all motifs are found with the highest frequency at the expected location, but that the DPE/MTE/PB motif is less specific and more frequently found at positions where it is unlikely to be functional as a core promoter element.”

*6) There are a confusing number of core promoter and MBT classification names. This is certainly not all the fault of the authors, but it would help the reader if the names from the traditional literature and the computational studies were both used or listed somewhere to help match Ohler and FitzGerald, for example. This could be added to the “promoter element table”. The authors should not send the reader to another paper for a term that they could easily define in the manuscript (dispersed and focused are never described). The long descriptors (“pre-MBT not-paused genes”) could be replaced in the figures with data in the heat maps. Almost anything that would make the high-dimensional naming of promoters easier to follow would be greatly appreciated*.

We also noticed that promoter studies and their classifications are not well known to a general audience. We have therefore now defined focused and dispersed promoters in the text, and mention their alternative names (peaked and broad promoters):

“Focused transcription initiates within a very narrow window and often at a single nucleotide (also called peaked promoters), while dispersed transcription initiates from several weak transcription start sites within a ∼150 nucleotide region (also called broad promoters). In *Drosophila*, dispersed initiation is typically found at broadly expressed housekeeping genes with constitutive promoters.”

We also evaluated whether we could simplify our nomenclature but only found one instance where it was not completely consistent. We had labeled the Ohler motifs as either Motif1, etc. or Ohler1, etc. We have now consistently referred to these motifs as Ohler1, etc. to make sure the source is clear (for example, as opposed to the FitzGerald nomenclature). Overall, we feel that our choice, description, and presentation of the core promoter elements is systematic:

• We only include core promoter elements in the analysis that are relatively well understood to avoid confusion.

• We are always using traditional names if available (DPE, DRE, GAGA, Inr, MTE, PB, TATA, Zelda) and use the Ohler nomenclature only if no other name is known (Ohler1, Ohler6, Ohler7).

• If possible, we always mention promoter elements that belong to the same promoter type together.

• In the figures, core promoter elements are sorted and grouped by function for clarity.